# Gibberellin-Mediated Sensitivity of Rice Roots to Aluminum Stress

**DOI:** 10.3390/plants13040543

**Published:** 2024-02-16

**Authors:** Long Lu, Xinyu Chen, Qinyan Tan, Wenqian Li, Yanyan Sun, Zaoli Zhang, Yuanyuan Song, Rensen Zeng

**Affiliations:** 1Key Laboratory of Ministry of Education for Genetics, Breeding and Multiple Utilization of Crops, College of Agriculture, Fujian Agriculture and Forestry University, Fuzhou 350002, China; lulong@fafu.edu.cn (L.L.); chenxinyu_fafu@163.com (X.C.); 13584189246@163.com (Q.T.); 19189360579@163.com (W.L.); yanyansun565340120@163.com (Y.S.); zzl0418ff@163.com (Z.Z.); 2Key Laboratory of Biological Breeding for Fujian and Taiwan Crops, Ministry of Agriculture and Rural Affairs, Fujian Agriculture and Forestry University, Fuzhou 350002, China

**Keywords:** aluminum toxicity, rice, gibberellin biosynthesis and signaling, ROS accumulation

## Abstract

Aluminum toxicity poses a significant constraint on crop production in acidic soils. While phytohormones are recognized for their pivotal role in mediating plant responses to aluminum stress, the specific involvement of gibberellin (GA) in regulating aluminum tolerance remains unexplored. In this study, we demonstrate that external GA exacerbates the inhibitory impact of aluminum stress on root growth of rice seedlings, concurrently promoting reactive oxygen species (ROS) accumulation. Furthermore, rice plants overexpressing the GA synthesis gene *SD1* exhibit enhanced sensitivity to aluminum stress. In contrast, the *slr1* gain-of-function mutant, characterized by impeded GA signaling, displays enhanced tolerance to aluminum stress, suggesting the negative regulatory role of GA in rice resistance to aluminum-induced toxicity. We also reveal that GA application suppresses the expression of crucial aluminum tolerance genes in rice, including *Al resistance transcription factor 1* (*ART1*), *Nramp aluminum transporter 1* (*OsNramp4*), and *Sensitive to Aluminum 1* (*SAL1*). Conversely, the *slr1* mutant exhibits up-regulated expression of these genes compared to the wild type. In summary, our results shed light on the inhibitory effect of GA in rice resistance to aluminum stress, contributing to a theoretical foundation for unraveling the intricate mechanisms of plant hormones in regulating aluminum tolerance.

## 1. Introduction

Aluminum, the most abundant metallic element in the earth’s crust, primarily exists as oxides or silicates [1]. Under soil conditions with a pH of ≤4.5, aluminum undergoes activation and dissolution into harmful Al^3+^ ions. These ions interact with negatively charged root cell walls, triggering an increase in reactive oxygen species (ROS) [2], impairing cell wall extensibility, and impeding root elongation [3,4]. Al^3+^ ions also adversely impact nutrient absorption, hindering overall plant growth [5]. Approximately 30% of global arable land is characterized by acidic soil, which poses a significant limitation to crop productivity due to aluminum toxicity [6,7].

Plants have evolved various mechanisms to tolerate aluminum stress [8]. They secrete organic acids to resist it [9,10], including malic acid [11,12] from *Brassica napus* and *Arabidopsis thaliana*, citric acid [13,14] from *Zea mays* and *Glycine max*, and oxalic acid from *Lycopersicon esculentum* [15]. The secretion of organic acids assists the chelation of aluminum ions to form complexes, reducing the toxic influences of aluminum on plants [16,17].

Additionally, plants are able to sequester aluminum ions into vacuoles to resist aluminum stress [18]. For example, in rice (*Oryza sativa*), the membrane-localized aluminum transporter protein OsNramp4 transports Al^3+^ from the root apoplast to the cytoplasm. Subsequently, Al^3+^ is sequestered into the vacuole through the activity of a tonoplast-localized half-size ABC transporter, OsALS1, which plays a crucial role in the internal detoxification of aluminum in rice [19,20]. Moreover, plasma membrane (PM) H^+^-ATPase activity may be regulated to influence plant aluminum absorption. This effect largely depends on the plant species [21,22].

Phytohormones, crucial signaling molecules in plant growth and development, also play a pivotal role in responding to environmental stresses [23]. Research indicates that plant hormones like abscisic acid [24], ethylene [25], and auxin [26] are involved in plant response to aluminum toxicity stress. External application of phytohormones and synthetic analogs can have similar effects [27]. Rice, known for its strong aluminum tolerance [28], possesses aluminum-resistance genes like the C_2_H_2_-type zinc finger transcription factor ART1, which regulates the expression of 31 downstream target genes, such as the resistance gene *NRAT1*/*OsNRAMP4* [29]. The ABA stress and ripening gene (ASR) family genes in rice are induced by abscisic acid and aluminum stress simultaneously. OsASR5, as a transcription factor, regulates the expression of three downstream aluminum tolerance genes *STAR1*, *NRAT1*, and *FRDL4* [30,31]. The application of jasmonic acid can enhance the growth inhibition of rice seedling roots caused by aluminum [32]. The auxin carrier (OsAUX3) in rice participates in aluminum-induced root growth inhibition [33]. Under aluminum stress, the induction of ethylene synthesis in wheat negatively regulates the *Aluminum-activated malate transporter 1* (*TaALMT1*) and reduces the root exudation of malic acid, making wheat more sensitive to aluminum toxicity stress [34].

Gibberellic acids (GAs) are a diverse group of tetracyclic diterpenoid carboxylic acids with the ability to stimulate seed germination, increase plant height, boost tiller number, and enhance fresh weight [35]. In higher plants, the GA biosynthesis pathway consists of three distinct stages. Copalyl diphosphate synthase (CPS) and ent-kaurene synthase (KS) play key roles in the initial two stages, primarily involved in the synthesis of GA precursors. Following cyclization, GA_12_ is formed in the endoplasmic reticulum through the catalytic action of ent-kaurene oxidase (KO) and ent-kaurenoic acid oxidase (KAO). In the third stage, GA_12_ is further metabolized in the cytoplasm by the dioxygenases GA20ox and GA3ox to yield bioactive gibberellins, such as GA_1_ and GA_4_. Additionally, GAs play a crucial role in modulating plant responses to various abiotic stresses. The function of GAs varies with different abiotic stresses. For example, under salt stress, gibberellin synthesis is suppressed to improve plant salt tolerance [36]. Conversely, when plants are submerged, the production of bioactive gibberellins increases to promote internode elongation and counteract the effects of waterlogging [37]. Similarly, gibberellin has multiple functions in response to metal stress. In the presence of copper stress, the level of bioactive gibberellin in tomato decreases [38]. GA is involved in the regulation of cell mitosis in broad beans, thereby mitigating the adverse effects of heavy metals on mitosis and metabolism in broad beans under cadmium and lead stress and positively affecting plant yield and seeding growth [39]. Gibberellins have been shown to have the potential to enhance plant resistance to heavy metals by increasing photosynthetic activity. Specifically, GAs have been found to improve the growth rate, chlorophyll content, and net CO_2_ assimilation rate of soybean plants when subjected to cadmium stress [40]. However, the role of GAs in coping with aluminum stress remains unexplored.

In this study, we investigated the impact of GA on aluminum tolerance in rice through external application. Our findings revealed that external GA intensified the growth inhibitory effects of aluminum stress on rice roots, promoted ROS accumulation, and down-regulated the expression of key aluminum tolerance genes. The *SD1*-OE transgenic rice (overexpressing the GA biosynthetic gene *SD1*) and the *slr1* gain-of-function mutant of *SLR1* (a key component of gibberellin signal transduction) were used to further identify the role of GA in rice aluminum tolerance. The *SD1*-OE transgenic rice exhibited similar responses to aluminum stress as external GA application, while the *slr1* mutant displayed enhanced tolerance to aluminum stress, unaffected by external GA. This study unveils GA’s potential negative regulation of GA in rice resistance to aluminum stress by influencing the expression of aluminum tolerance genes and ROS accumulation in roots.

## 2. Results

### 2.1. Expression Patterns of GA Synthesis and Signal Transduction Genes in Response to Aluminum Stress

To investigate the impact of aluminum stress on genes associated with GA synthesis and signal transduction, we assessed the expression of GA biosynthesis-related genes and the key signal transduction component, *SLR1*, under aluminum stress. In rice roots exposed to aluminum stress, distinct expression patterns were observed for various GA synthesis genes. Specifically, *SD1* (*OsGA20ox2*) exhibited a significant upregulation, which upregulated at 3 and 6 h, and then gradually decreased to the initial level after 12 h. The expression of GA degradation-related genes displayed complexity: *OsGA2ox3* peaked at 3 h, *OsGA2ox6* and *OsGA2ox9* were significantly upregulated at 3 and 12 h after Al treatment, whereas *OsGA2ox5* was downregulated after Al treatment (Figure 1).

Furthermore, the expression level of *SLR1*, a negative regulator of GA signal transduction, peaked after 3 h of aluminum stress but was lower than the control at 24 h. (Figure 1). These findings highlight the dynamic impact of aluminum stress on the expression of genes involved in GA synthesis and signal transduction. The observed changes suggest the potential involvement of GA in regulating aluminum tolerance in rice.

### 2.2. External Application of GA Exacerbates the Inhibitory Effect of Root Growth by Aluminum

To elucidate the role of GA under aluminum stress, we investigated the impact of externally applied GA at various concentrations on rice growth under aluminum stress (Figure 2A). As the inhibition of root growth is the most immediate and prominent phenotype in plants under aluminum stress, the relative root length (RPL) serves as a crucial indicator for assessing aluminum toxicity [41]. Consequently, we measured the relative length of rice roots exposed to different concentrations of GA (Figure 2B).

Our findings demonstrated that GA promoted root elongation in wild-type rice; however, it exacerbated the inhibitory effect of aluminum on rice root growth under aluminum stress. The inhibition rate increased proportionally with the concentration of GA, peaking at 50% inhibition (Figure 2B). This trend was further confirmed by the determination of root fresh weight, reflecting that under control conditions, external GA application enhanced the growth and biomass accumulation of rice roots. Conversely, under aluminum stress, external GA application intensified the inhibitory impact of aluminum on root growth (Figure 2C).

### 2.3. Overexpression of the GA Biosynthesis Gene SD1 Negatively Modulates Rice Resistance to Aluminum Stress

To elucidate the influence of GA biosynthesis in rice on aluminum tolerance, we analyzed the phenotypes of rice *SD1*-OE seedlings overexpressing the GA biosynthesis gene *SD1* under aluminum stress (Figure 3A). Under normal conditions, the roots of *SD1*-OE seedlings were notably longer than those of the wild-type Nip. However, under aluminum stress, there was no discernible difference in root length between *SD1*-OE and Nip. Analysis of the relative root length revealed that under aluminum stress, the relative root length of *SD1*-OE was significantly lower than that of Nip (Figure 3B). Moreover, the biomass accumulation of *SD1*-OE exhibited a more substantial decrease compared to Nip, indicating that SD1-OE transgenic rice is more susceptible to aluminum stress than the wild-type (Figure 3C).

### 2.4. Disruption of the GA Signaling Enhances Rice Aluminum Resistance

To further investigate the involvement of GA signaling in rice aluminum tolerance, we utilized *slr1* plants as the experimental material, analyzing their performance under aluminum stress with the addition of exogenous GA or under control conditions, in comparison with the wild-type plants (Figure 4A). The results indicate that under normal conditions, GA fails to promote *slr1* growth, suggesting an effective blockade of GA signaling in *slr1*. Assessment of the relative root length revealed that under aluminum stress, exogenous GA intensified the relative growth inhibition of wild-type rice 9311 roots. Whereas *slr1* exhibited insensitivity to aluminum stress, displaying significantly better relative root length than wild-type 9311 (Figure 4B). Moreover, the application of GA had no discernible effect on the growth of *slr1* under aluminum stress. Under aluminum stress, the root biomass of *slr1* was significantly higher than that of the wild type, and the external application of GA had no significant impact on the biomass accumulation of *slr1* (Figure 4C). These findings suggest that blocking GA signaling enhances the resistance of rice seedlings to aluminum stress.

### 2.5. GA Promotes ROS Accumulation in Rice Roots under Aluminum Stress

Amid aluminum stress, the presence of Al^3+^ induces the significant accumulation of ROS in rice root tips, resulting in lipid peroxidation and ultimately affecting root growth. [42]. To assess ROS accumulation, rice root tips were stained using nitroblue tetrazolium (NBT) and 3,3′-diaminobenzidine (DAB). Our observations revealed that exogenous GA led to increased ROS accumulation in wild-type rice root tips, whereas *slr1* exhibited reduced ROS accumulation (Figure 5A,B). Elevated ROS levels induce peroxidation of plant membrane lipids, generating malondialdehyde (MDA) as a byproduct. MDA levels serve as an indicator of cell oxidation levels. Subsequent testing of MDA content in rice roots indicated that GA application increased MDA accumulation in aluminum-stressed wild-type rice, while *slr1* showed no significant difference (Figure 5C). These results suggest that blocking GA signaling can mitigate the toxic effects of aluminum stress on rice roots by reducing ROS accumulation.

### 2.6. GA Influences the Expression of Aluminum Tolerance Genes in Rice

To unravel the molecular mechanism through which GA heightens aluminum sensitivity in rice, we investigated the impact of GA application on the expression of aluminum tolerance genes in wild-type rice roots under both control and aluminum stress conditions. As illustrated in Figure 6, GA notably diminished the expression of *Nramp aluminum transporter 1* (*NRAT1*/*OsNramp4*), *Al resistance transcription factor 1* (*ART1*), and *Sensitive to Aluminum 1* (*SAL1*) in rice roots. In comparison to the wild type, the expression levels of these genes were lower in the roots of *SD1*-OE transgenic rice, while they were significantly up-regulated in the roots of *slr1*. These results suggest that GA may influence rice root growth and ROS accumulation under aluminum stress by modulating the expression of aluminum resistance-related genes.

## 3. Discussion

Gibberellins are an important class of phytohormones with a range of roles in regulating plant growth, development, and stress response. However, the role of GA in plant response to aluminum stress remains poorly understood. Here, we reveal that GA negatively modulates aluminum stress tolerance in rice through external GA application and phenotype analysis of *SD1*-OE transgenic plants and the *slr1* gain-of-function mutant. These results provide new insights into the regulation of GA in Al tolerance in rice.

*SLR1* encodes the DELLA protein in rice, functioning as a negative regulator in GA signal transduction. Upon reaching a certain concentration, GA promotes SLR1 protein degradation, facilitating GA signal transmission. This reciprocal modulation between GA and SLR1 involves a dynamic interplay [43], whereby GA stimulates the transcription and translation of SLR1, leading to feedback inhibition of GA signaling. Typically, GA is recognized for its role in alleviating growth inhibition by disrupting DELLA protein function, thus promoting overall plant growth and development [44]. However, studies suggest that the accumulation of DELLA proteins can influence ROS accumulation, contributing to enhanced plant tolerance [45]. GA and DELLA proteins collaborate to orchestrate plant responses to various abiotic stresses [46]. For instance, under low-temperature stress in *Arabidopsis*, GA biosynthesis gene expression is down-regulated, resulting in increased DELLA protein accumulation, subsequently regulating downstream transcription factors and improving tolerance to low temperature [47]. Despite this, limited reports explore the role of GA signaling in regulating plant tolerance to heavy metal stress.

This study reveals that under aluminum stress, *SLR1* expression in rice roots is induced by aluminum signals (Figure 1). Notably, the *slr1* gain-of-function mutant exhibits heightened aluminum tolerance compared to the wild-type under aluminum stress. Conversely, wild-type rice treated with GA shows more pronounced root growth inhibition (Figure 4). This inhibitory effect intensifies with higher GA concentrations (Figure 2). Similarly, *SD1*-OE transgenic rice seedlings demonstrate increased sensitivity to aluminum stress (Figure 3). Examination of MDA and ROS levels indicated that the *slr1*-gain mutant exhibited decreased ROS accumulation and lower MDA content compared to the wild type when subjected to aluminum stress. In contrast, rice seedlings treated with GA displayed increased ROS accumulation and higher MDA content (Figure 5). These findings suggest that inhibiting GA signaling through SLR1 mitigates root growth inhibition by reducing ROS levels, while GA relieves this inhibition, thereby negatively regulating aluminum tolerance in rice. Additionally, SLR1 acts not only as a pivotal factor in GA signaling but also as a central node in the signal interaction of other plant hormones. Existing reports indicate that SLR1 specifically interacts with the jasmonic acid signaling pathway transcription factor MYC2 [48]. Given that MYC2 is involved in regulating tomato root growth under aluminum stress [49], it suggests a potential interaction between GA and other plant hormones in responding to aluminum stress.

ART1 is recognized as a pivotal transcription factor involved in the regulation of aluminum tolerance in rice. Constitutively expressed in rice roots, it acts as an upstream regulatory protein for multiple key aluminum tolerance genes, including *OsSTAT1* and *OsSTAR2*, which encode an ATP-binding and a transmembrane domain of a bacterial-type ATP-binding cassette transporter, respectively. Knockout of either gene resulted in sensitivity to Al toxicity [50]. ART1 downstream targets play crucial roles in mitigating aluminum toxicity at various cellular levels. These processes involve the sequestration of Al^3+^ through transporters in rice and the release of organic anions from rice roots to form non-toxic chelates with Al^3+^. Two transporters essential for Al tolerance, OsNramp4 and OsALS1, are regulated by ART1. OsNramp4 functions as a key transporter protein for the translocation of Al into the cytoplasm, while OsALS1 is responsible for sequestering Al into the vacuole [19]. Additionally, SAL1, known for regulating malic acid secretion in *Arabidopsis* to influence aluminum tolerance, exhibits a distinct role in rice by inhibiting plasma membrane H^+^-ATPase activity through the process of dephosphorylation to regulate aluminum uptake [51]. Our study unveils that the expression of *ART1*, *OsNramp4*, and *SAL1* is up-regulated in *slr1*, indicating enhanced aluminum tolerance. In contrast, *SD1*-OE rice shows down-regulation of these genes, suggesting an increased sensitivity to aluminum stress. External application of GA also leads to the down-regulation of these three genes. This implies that GA may exert a negative regulatory influence on the expression of rice-related aluminum tolerance genes, influencing the absorption and transport of aluminum in rice by affecting *ART1*. In addition, the activation of ART1 is modulated by protein phosphorylation or its interaction with other transcription factors, while its expression level is not affected by Al treatment [29]. Interestingly, in the *slr1* mutant, the expression of *ART1* is found to be further increased under aluminum stress (Figure 6). These results suggest that SLR1, a key regulator in the gibberellin signaling pathway, is highly likely to have a direct or indirect impact on the expression of ART1.

In summary, our investigation reveals that aluminum stress impacts the dynamic expression of genes associated with GA synthesis and signal transduction. Furthermore, GA exacerbates the inhibition of rice seedling root growth by aluminum and concurrently suppresses the expression of aluminum-tolerant genes in rice. This establishes a negative regulatory function of GA in aluminum tolerance in rice seedlings. Additionally, our findings suggest that SLR1 may serve as a key factor in the regulation of aluminum tolerance in rice. SLR1 modulates rice tolerance to aluminum stress by influencing the expression of downstream aluminum tolerance-related genes. Consequently, our study sheds light on the crucial role of GA signal transduction in the regulation of aluminum tolerance in rice, offering a new theoretical foundation and technical support for enhancing crop traits related to aluminum toxicity through molecular genetic improvement in the future.

## 4. Materials and Methods

### 4.1. Plant Materials

The wild-type rice utilized in this experiment was Indica rice “9311” and Japonica rice “Nip.” The transgenic *SD1*-overexpressing transgenic rice plants (*SD1*-OE) and mutant *slr1* were previously described and generously provided by Professor Dabing Zhang from Shanghai Jiao Tong University [52]. The *SD1* gene encodes gibberellin 20-oxidase, which controls rice plant height. *SD1*-OE has a higher plant height than wild-type Nip. SLR1 is the repressor of Gibberellin signaling. The *slr1* gain-of-function mutant contains a mutation in the GA-perceived domain and exhibits a dwarf phenotype. Rice seeds were disinfected with 2% (*v*/*v*) sodium hypochlorite (NaClO) for 15 min, followed by rinsing with distilled water 5–7 times. The seeds were then transferred to conical flasks, and germinated at 28 °C for 48 h. The germinated seeds were transferred to a cultivation box. All rice seedlings were grown in a greenhouse at 28 °C under a photoperiod of 14 h of light/10 h of darkness.

### 4.2. Aluminum Resistance Testing

As previously described [51], five-day-old hydroponic seedlings were transferred to 96 well plates floating on a 2 L plastic box (L × W × H: 22 cm × 14 cm × 7 cm) with 0.5 mM CaCl_2_ solution (pH 4.5) containing 50 μM AlCl_3_ and grown for 7 days. After this treatment, the seedling phenotype was measured. The relative root length of seedling roots was assessed using a ruler, calculated as the percentage of the root length treated with Al relative to the root length untreated with Al. while the fresh weight was determined by an electronic scale to assess aluminum tolerance.

### 4.3. Exogenous Gibberellin Application

To investigate the influence of GA on the aluminum tolerance of rice seedlings, 5-day-old wild-type, and *slr1* mutant rice seeds were transferred to 0.5 mM CaCl_2_ solutions (pH of 4.5) containing 0 μM or 50 μM AlCl_3_, as well as 0, 1, 2, and 5 μM GA, for 7 days. The GA was purchased from Aladdin Biochemical Company (Shanghai, China) and then solubilized in anhydrous ethanol to create a 10 mM concentrated solution. Following this treatment, the phenotype, encompassing relative root length and fresh weight of the seedlings, was detected.

### 4.4. RNA Extraction

To elucidate the effects of external GA application, as well as GA biosynthesis and signal transduction in plants, on the expression of aluminum tolerance genes in the roots of rice seedlings, five-day-old rice seedlings were transferred to a 0.5 mM CaCl_2_ solution (pH of 4.5) supplemented with 0 μM or 50 μM AlCl_3_ and 0 μM or 1 μM GA. Following 6 h of treatment, the roots were removed to isolate RNA. Total RNA was extracted utilizing TRNzol universal reagent (Tiangen, Beijing, China) following the manufacturer’s instructions. All samples were stored at −80 °C for gene expression analysis.

### 4.5. Real-Time Fluorescence Quantitative PCR Analysis (RT-qPCR)

First-strand cDNA was generated using a first-strand Synthesis Master Mix (Lablead, Beijing, China). RT-qPCR analysis was conducted using a StepOnePlus real-time PCR instrument (Applied Biosystems, Waltham, MA, USA) with 2 × Realab Green PCR Fast Mix (Lablead, Beijing, China) to identify gene expression. *OsACTIN* was utilized as the internal reference gene. Fluorescence measurements were collected, and amplicon specificity was confirmed by performing melting curve analysis and agarose gel electrophoresis. The transcriptional abundance of the gene of interest was determined using the double standard curve approach. Reaction conditions for thermal cycling were initial denaturation at 95 °C for 30 s, 40 cycles of denaturation for 10 s at 95 °C, and extension for 30 s at 60 °C. All RT-qPCR experiments included three independent samples for analysis. Sequence information about primers utilized for gene expression analysis is denoted in Appendix A.

### 4.6. MDA Examination

Five-day-old rice seedlings were moved to a 0.5 mM CaCl_2_ solution (pH 4.5) containing 0 μM or 50 μM AlCl_3_ and 0 μM or 1 μM GA. Following 7 days of treatment, the roots were detached to characterize malondialdehyde (MDA). All these measurements were biologically repeated three times. Root samples (0.1 g) were collected and ground in liquid nitrogen. They were extracted with 0.05 mM PBS buffer (10 mL, pH 7.8). The resulting mixture was transferred to a centrifuge tube and centrifuged at 6000× *g* rpm for 20 min. The resulting supernatant was stored at 4 °C for further use. The use of thiobarbituric acid (TBA) to detect MDA content has been previously reported [53].

### 4.7. ROS Accumulation Measurement

Five-day-old rice seedlings were subjected to aluminum stress for 5 days, after which the underground portions were immediately collected for the identification of ROS accumulation. Following established protocols, the superoxide anion (O_2_^−^) was analyzed by staining the roots in the dark for 15 min with 6 mM NBT in 50 mM PBS buffer (pH 7.5) [54]. Hydrogen peroxide (H_2_O_2_) content was determined by the DAB staining method [55]. Briefly, the roots were stained for 30 min in the dark using 1 mg/mL DAB and 0.05% (*v*/*v*) Tween 20 in 10 mM Na_2_HPO_4_. The stained roots were observed and photographed using a stereomicroscope (Nikon, SMZ18, Tokyo, Japan) equipped with a camera (Nikon, DS-U3, Tokyo, Japan).

### 4.8. Statistical Analysis

SPSS statistics software (Version 26.0 for Windows, SPSS, Chicago, IL, USA) was utilized to characterize the differences of all experimental data. Student *t*-tests were used to analyze relative root length, fresh weight, and gene expression significance. For the analysis of MDA content, a one-way ANOVA was performed, followed by a Tukey test to determine the significance.

## Figures and Tables

**Figure 1 plants-13-00543-f001:**
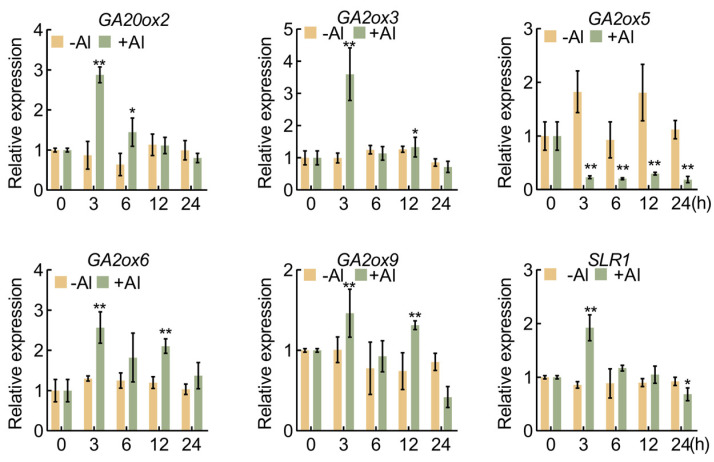
Expression of genes involved in gibberellin biosynthesis and signal transduction under aluminum stress. Quantification of GA biosynthesis genes *GAs* and GA signaling gene *SLR1* upon Al treatment. Error bars indicate SD (n = 3). Asterisks indicate significant differences compared with the corresponding controls (* *p* < 0.05; ** *p* < 0.01). The experiments were independently repeated at least three times.

**Figure 2 plants-13-00543-f002:**
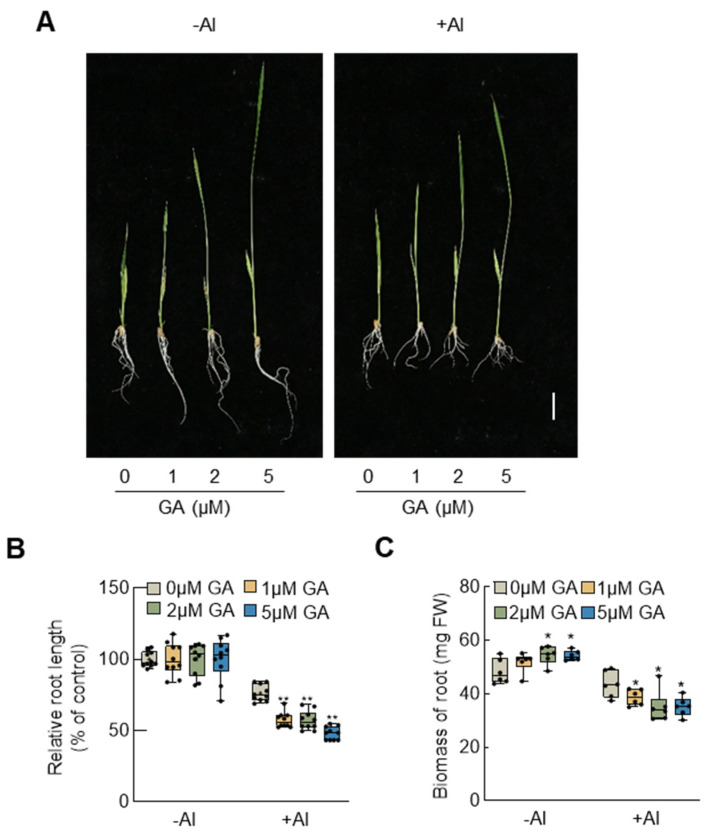
Influence of gibberellin on the phenotype of rice seedlings subjected to aluminum stress. (**A**) Performance of 5-day-old rice seedlings treated with gibberellin at different concentrations for 7 days in the presence of Al stress. Scale bar = 5 cm. (**B**) Relative root length of rice seedlings treated with water or different concentrations of gibberellin under Al stress. Error bars indicate SD (n = 10). (**C**) Biomass of roots of rice seedlings treated with water or different concentrations of gibberellin under Al stress conditions. Error bars indicate SD (n = 6). Asterisks indicate significant differences compared with the corresponding controls (* *p* < 0.05; ** *p* < 0.01). The experiments were independently repeated at least three times.

**Figure 3 plants-13-00543-f003:**
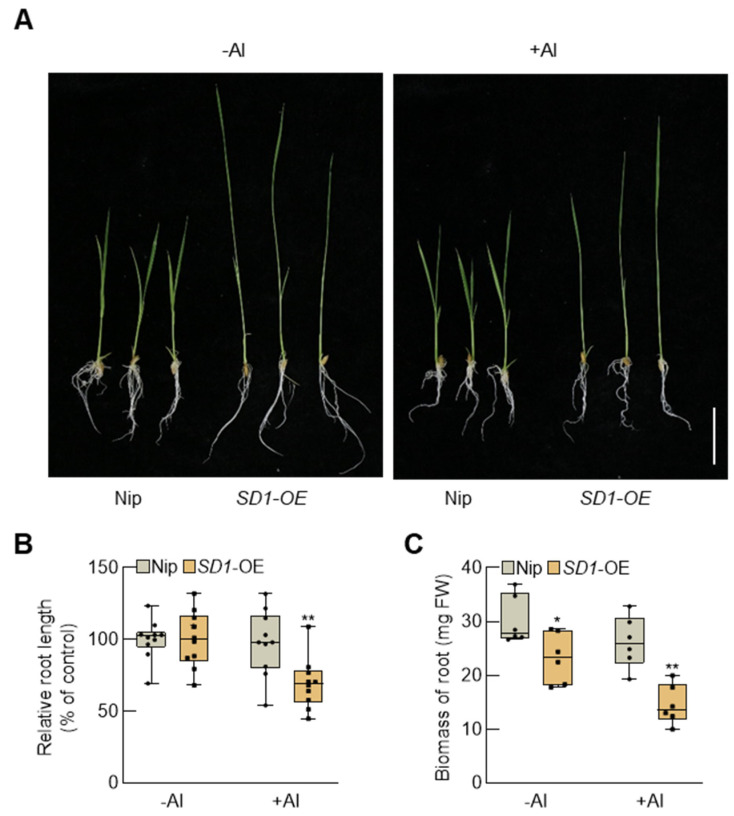
Overexpression of the *SD1* negatively regulates rice resistance to aluminum stress. (**A**) Performance of 5-day-old wild-type and *SD1*-OE seedlings under Al stress conditions for 7 days. Scale bar = 5 cm. (**B**) Relative root length of the wild-type and *SD1*-OE seedlings after Al treatment. Error bars indicate SD (n = 10). (**C**) Biomass of roots of the wild-type and *SD1*-OE seedlings under Al stress conditions. Error bars indicate SD (n = 6). Asterisks indicate significant differences compared with the corresponding controls (* *p* < 0.05; ** *p* < 0.01). The experiments were independently repeated at least three times.

**Figure 4 plants-13-00543-f004:**
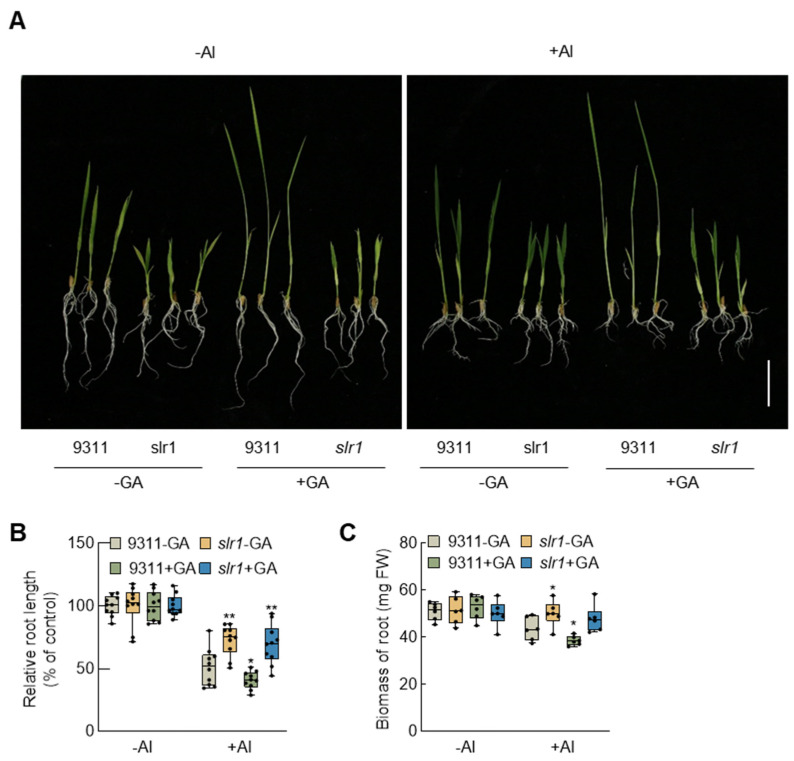
Blocking gibberellin signaling enhances aluminum resistance in rice seedlings. (**A**) Performance of 5-day-old wild-type and *slr1* seedlings after GA treatment in the presence of Al toxicity for 7 days. Scale bar = 5 cm. (**B**) Effect of external GA treatment on the relative root length of the wild-type and *slr1* seedlings under Al stress conditions. Error bars indicate SD (n = 10). (**C**) Effect of GA application on root biomass of the wild-type and *slr1* seedlings under Al stress. Error bars indicate SD (n = 6). Asterisks indicate significant differences compared with the corresponding controls (* *p* < 0.05; ** *p* < 0.01). The experiments were independently repeated at least three times.

**Figure 5 plants-13-00543-f005:**
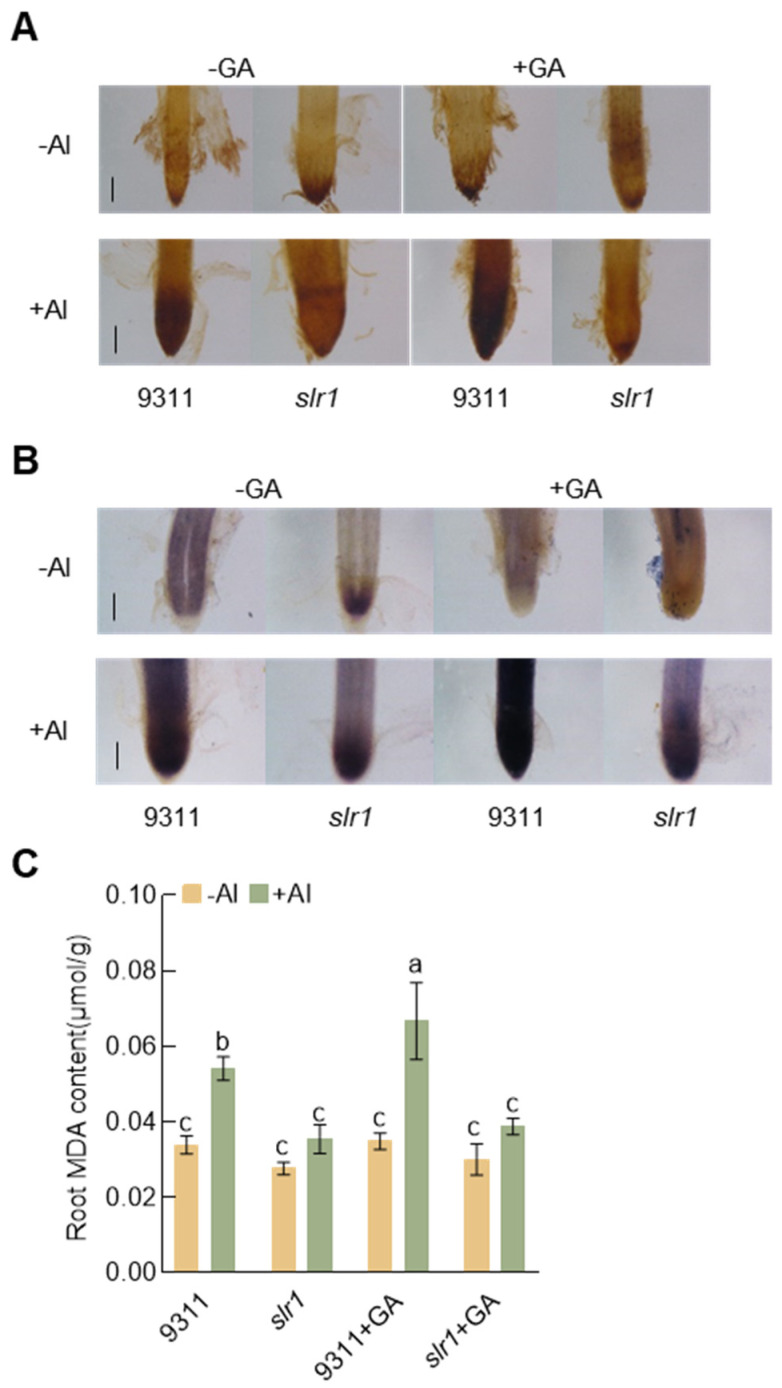
Gibberellin exacerbates the accumulation of ROS in rice under Al stress. (**A**) DAB staining of roots of the wild-type and *slr1* seedlings with or without GA treatment under Al stress conditions. The brown color indicates the H_2_O_2_ level in each root. (**B**) NBT staining of roots of the wild-type and *slr1* seedlings with or without GA treatment under Al stress. The blue color indicates the O_2_^−^ level in each root. (**C**) Measurement of the MDA contents in roots of the wild-type and *slr1* seedlings with or without GA treatment under Al stress conditions. Error bars indicate SD (n = 3). The lowercase letters ‘a’ to ‘c’ indicate significant differences as determined by Tukey’s test (*p* < 0.05 or *p* < 0.01). Scale bar = 200 μm (**A**,**B**). The experiments were independently repeated at least three times.

**Figure 6 plants-13-00543-f006:**
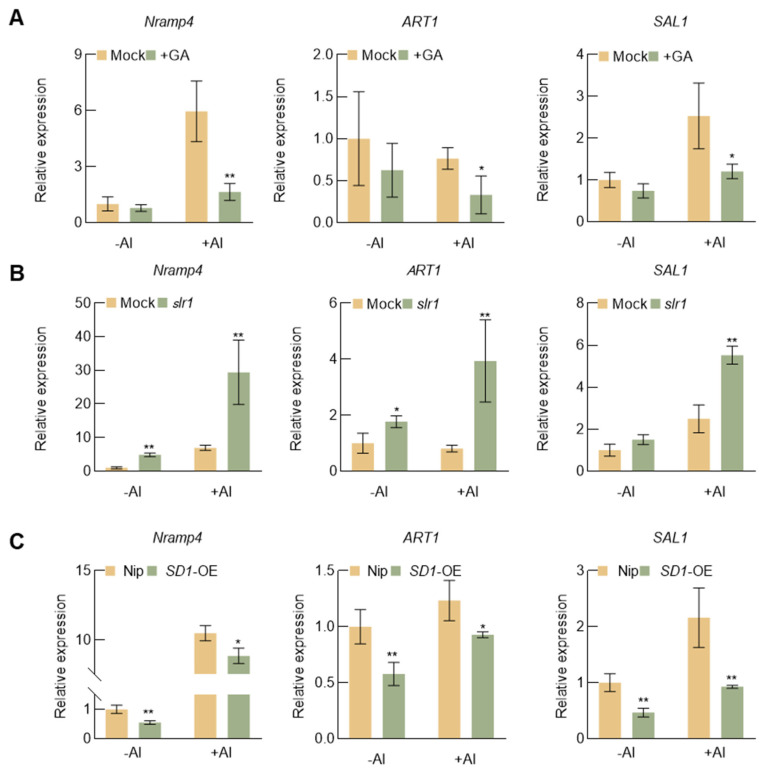
Effect of gibberellin on the expression of genes related to aluminum tolerance. (**A**) Influence of GA application on the expression of *OsNramp4*, *ART1*, and *SAL1* in wild-type roots under Al stress conditions. Error bars indicate SD (n = 3). (**B**) Transcript levels of *OsNramp4*, *ART1*, and *SAL1* in wild-type and *slr1* roots under Al stress conditions. Error bars indicate SD (n = 3). (**C**) Transcript levels of *OsNramp4*, *ART1*, and *SAL1* in wild-type and *SD1*-OE roots under Al stress conditions. Error bars indicate SD (n = 3). Asterisks indicate significant differences compared with the corresponding controls (* *p* < 0.05; ** *p* < 0.01). All experiments were repeated independently three times.

## Data Availability

All the data analyzed during this study have been included in this article.

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
