# Peer review of "Gibberellin-Mediated Sensitivity of Rice Roots to Aluminum Stress"

_plants, 2024, doi:10.3390/plants13040543_

Round 1

Reviewer 1 Report

Comments and Suggestions for Authors

The study on the effect of GA in rice resistance to aluminum stress is interesting.

Some minor comments:

Line 123 and Figure 2A, the shoot growth of the seedlings was also influenced by Al and GA, is there any explanation?

Figure 5 A and B, please add scale bar. The quality of Figure 5 B should be improved.

Author Response

1.Line 123 and Figure 2A, the shoot growth of the seedlings was also influenced by Al and GA, is there any explanation?

Response: Research on aluminum tolerance in rice has mainly focused on the effect of aluminum on the growth of rice roots, since the primary and earliest symptom of Al toxicity is a rapid (beginning within minutes) inhibition of root growth (Kochian et al, 2005, DOI: 10.1007/s11104-004-1158-7; Wang et al, 2014, DOI: 10.1016/j.jprot.2013.12.023; Huang et al, 2023, DOI: 10.1093/plcell/koad281). Therefore, we investigated the effect of Al and GA on rice roots. The application of 1, 2, and 5 μM GA improved shoot growth but inhibited root growth under Al stress conditions when compared to the control. There was no difference in plant height between seedlings with or without GA application, except those with 5 μM GA, under normal and aluminum stress conditions. However, root growth was significantly inhibited in all groups of seedlings under aluminum stress conditions, and the application of GA exacerbated this inhibition. These results suggest that roots are more sensitive than shoots under aluminum stress and that GA enhances the inhibitory effect of aluminum on root growth.

2. Figure 5 A and B, please add scale bar. The quality of Figure 5 B should be improved.

Response: We have included scale bars in Figure 5A and Figure 5B, and provided updated Figure 5B in the revised manuscript.

Reviewer 2 Report

Comments and Suggestions for Authors

Authors have provided very interesting results regarding the interaction of hormonal regulation and aluminum toxicity in rice plants. The work is well written and deserves publication. I have a few comments regarding the clarity of some aspects:

1.     In the introduction (line 45), the authors indicate that OsNramp4 is an Al3+ transporter, suggesting its role in vacuole Al3+ sequestration. However, Hao et al. 2022 (https://doi.org/10.1016/j.repbre.2022.10.001) clearly showed that OsNramp4 is localized in the plasma membrane (also responsible for transporting Cd2+). Therefore, OsNramp4 transports Al3+ (and Cd2+) into the root cell cytoplasm, not the vacuole. Aluminum is further sequestered into a vacuole by other vacuolar transporters, e.g., OsALS1 or others. It is important to note that Nramps were shown to mediate ion transport to the cytoplasm. If OsNramp4 would be localized in the vacuolar membrane, it would transport Al3+ out of the vacuole (similar to Fe3+ stored in the vacuole being released by AtNRAMP4/AtNRAMP3 localized in the vacuolar membrane) (Lanquar et al. 2005; 10.1038/sj.emboj.7600864).

2.     In the results section (part 2.1) titled "Expression Patterns of GA Synthesis,(…)" there is a need to put the results in perspective. Assuming the photoperiod was 14d/10n, I assume that samples were taken only during the day(?). If so, this should be indicated on the figure, for example, by an inset showing a white bar when the light is on and a dark one when it is off. Most importantly, authors should consider the impact of circadian or diurnal rhythms on the results, especially when comparing expression to time 0. Authors must clarify whether the control involves independent samples (not treated with Al) compared to samples at the same times after Al treatment (e.g., Al non-treated at 3h compared to Al treated after 3h) or if all samples are only compared to time 0. If the latter is true, the interpretation of the results in 2.1 becomes speculative and complex. GA signaling is strongly dependent on day/night, and comparing all samples to time 0 which is rhythm transition time (I assume sample was taken around hour after lights on) may introduce bias. If that is the case authors must indicate all of those potential issues and explain consequences – this will let readers have certain distance to interpret those results.

3.     Alternatively, considering a more accurate control (if Al non-treated are compared to Al treated at a given time), the patterns of expression of the GA synthesis pathway seem quite similar and could be described as upregulated (GA2ox2, 3, 6, and 9) starting at 3h after treatment, with expression diminishing after 12h (possibly due to circadian rhythms). The potential downregulation after 24h for GA2ox3 and GA2ox9 might be related to mechanisms suggesting the importance of GA in response to persistent Al stress. This hypothesis is supported by SLR1.

Anyway, the statement "dynamic impact" is currently too general and should be explained in more detail.

4.     In lines 266-267, please add information about the physiological function of OsSTAT1 and OsSTAR2.

5.     In lines 270-272, as in the introduction, OsNramp4 was shown to have localization in the plasma membrane and therefore cannot be directly responsible for vacuolar sequestration. In respect to the role of OsNramp4 in Al tolerance, authors should indicate that an additional transporter from cytoplasm to vacuole is needed. Its expression should be correlated with OsNramp4.

6.     In lines 285-287: "These results suggest that SLR1, a key regulator in the gibberellin signaling pathway, is highly likely to have a direct or indirect impact on the expression of ART1." Authors should point out (refer) what would involve an indirect pathway to regulate ART1.

Author Response

  1. In the introduction (line 45), the authors indicate that OsNramp4 is an Al3+transporter, suggesting its role in vacuole Al3+ sequestration. However, Hao et al. 2022 (https://doi.org/10.1016/j.repbre.2022.10.001) clearly showed that OsNramp4 is localized in the plasma membrane (also responsible for transporting Cd2+). Therefore, OsNramp4 transports Al3+ (and Cd2+) into the root cell cytoplasm, not the vacuole. Aluminum is further sequestered into a vacuole by other vacuolar transporters, e.g., OsALS1 or others. It is important to note that Nramps were shown to mediate ion transport to the cytoplasm. If OsNramp4 would be localized in the vacuolar membrane, it would transport Al3+ out of the vacuole (similar to Fe3+ stored in the vacuole being released by AtNRAMP4/AtNRAMP3 localized in the vacuolar membrane) (Lanquar et al. 2005; 10.1038/sj.emboj.7600864).

Response: Sorry for the confusion caused by our inaccurate description of OsNarmp4. OsNrmap4 is a membrane-localized aluminum transporter protein that transports Al from the root apoplast to the cytoplasm. Then, Al is sequestered into the vacuole by a tonoplast-localized half-size ABC transporter, OsALS1, which is required for internal detoxification of Al in rice. We have rewritten the sentence in the revised manuscript (line45-49).

  1. In the results section (part 2.1) titled "Expression Patterns of GA Synthesis,(…)" there is a need to put the results in perspective. Assuming the photoperiod was 14d/10n, I assume that samples were taken only during the day(?). If so, this should be indicated on the figure, for example, by an inset showing a white bar when the light is on and a dark one when it is off. Most importantly, authors should consider the impact of circadian or diurnal rhythms on the results, especially when comparing expression to time 0. Authors must clarify whether the control involves independent samples (not treated with Al) compared to samples at the same times after Al treatment (e.g., Al non-treated at 3h compared to Al treated after 3h) or if all samples are only compared to time 0. If the latter is true, the interpretation of the results in 2.1 becomes speculative and complex. GA signaling is strongly dependent on day/night, and comparing all samples to time 0 which is rhythm transition time (I assume sample was taken around hour after lights on) may introduce bias. If that is the case authors must indicate all of those potential issues and explain consequences – this will let readers have certain distance to interpret those results.

Response: To exclude the effects of circadian rhythms, we analyzed the expression of GA biosynthesis signaling genes under normal conditions and Al stress. The results have been included in the revised manuscript (Fig.1, line109-116).

  1. Alternatively, considering a more accurate control (if Al non-treated are compared to Al treated at a given time), the patterns of expression of the GA synthesis pathway seem quite similar and could be described as upregulated (GA2ox2, 3, 6, and 9) starting at 3h after treatment, with expression diminishing after 12h (possibly due to circadian rhythms). The potential downregulation after 24h for GA2ox3 and GA2ox9 might be related to mechanisms suggesting the importance of GA in response to persistent Al stress. This hypothesis is supported by SLR1.

Anyway, the statement "dynamic impact" is currently too general and should be explained in more detail.

Response: We have supplemented Al non-treated as the control and provided a more detail description of gene expression patterns in the revised manuscript (line109-116).

  1. In lines 266-267, please add information about the physiological function of OsSTAT1 and OsSTAR2.

Response: STAT1 and STAR2 encode an ATP-binding and a transmembrane domain of a bacterial-type ATP-binding cassette transporter, respectively. Knockout of either gene resulted in sensitivity to Al toxicity. We have included the description of STAT1 and STAR2 in the revised manuscript (line 277-279).

  1. In lines 270-272, as in the introduction, OsNramp4 was shown to have localization in the plasma membrane and therefore cannot be directly responsible for vacuolar sequestration. In respect to the role of OsNramp4 in Al tolerance, authors should indicate that an additional transporter from cytoplasm to vacuole is needed. Its expression should be correlated with OsNramp4.

Response: We have rewritten the sentence in the revised manuscript (line282-285).

  1. In lines 285-287: "These results suggest that SLR1, a key regulator in the gibberellin signaling pathway, is highly likely to have a direct or indirect impact on the expression of ART1."Authors should point out (refer) what would involve an indirect pathway to regulate ART1.

Response: Thank you for your suggestion. Currently, there are no reports on the genes or pathways that affect ART1 expression in rice. 

Reviewer 3 Report

Comments and Suggestions for Authors

Manuscript Number: Plants 2023, 12, x. https://doi.org/10.3390/xxxxx

Title: Gibberellin-Mediated Sensitivity of Rice Roots to Aluminum Stress

Authors: Long Lu, Xinyu Chen, Qinyan Tan, Wenqian Li, Yanyan Sun, Zaoli Zhang, Yuanyuan Song, Rensen Zeng

In this manuscript, the authors investigated the impact of gibberellic acid (GA) on aluminium tolerance in rice. In experiments, they utilized wild-type Indica rice "9311" and Japonica rice "Nip", and transgenic SD1-OE and slr1 mutant. The experiments included:

  1. Assessment of the relative root length and biomass of four rice genotypes exposed to different concentrations of GA and aluminum.
  2. Expression analysis of GA biosynthesis-related genes and the key signal transduction component, SLR1, under aluminum stress.
  3. Assessment of ROS and hydrogen peroxide (H2O2) accumulation in rice under Al stress.
  4. Expression analysis of OsNramp4, ART1, and SAL1 in roots of wild-type rice, SD1-OE and slr1 mutant under Al stress conditions.

The results showed that under aluminum stress, exogenous GA led to increased ROS accumulation in root tips of wild-type rice '9311', contrary to slr1, which exhibited reduced ROS accumulation. Moreover, GA intensified the relative growth inhibition of wild-type rice 9311 roots compared to slr1, which displayed significantly better relative root growth than wild-type 9311. GA application suppresses the expression of crucial aluminum tolerance genes in roots of wild-type rice. Compared to the wild-type, the expression levels of these genes were lower in the roots of SD1-OE transgenic rice, while they were significantly up-regulated in the roots of slr1

The experiments were well-planned and documented. Results were statistically elaborated based on the three repetitions. Quality of written English is suitable for publication.

Minor remarks:

1) Please check the name Sensitive to Aluminum 1 – the letter 'i' is missing

2) Support a more detailed description of the tested genotypes in 4.1 Plant Materials.

What is caused by the introduced mutations/transformations in genotypes SD1-OE and slr1? Whether the response of wild genotypes to aluminum was known from previous experiments?

Author Response

1. Please check the name Sensitive to Aluminum 1– the letter 'i' is missing

Response: Thank you for your suggestion., the spelling mistakes have been corrected in the revised manuscript (line23, line220-221).

2. Support a more detailed description of the tested genotypes in 4.1 Plant Materials.

What is caused by the introduced mutations/transformations in genotypes SD1-OE and slr1? Whether the response of wild genotypes to aluminum was known from previous experiments?

Response: The SD1 gene encodes gibberellin 20-oxidase that controls rice plant height. SD1-overexpressing transgenic rice plants (SD1-OE) have higher plant height than wild-type Nip. SLR1 is the repressor of Gibberellin signaling. slr1 gain-of-function mutant contains a mutation in GA-perceived domain and exhibits a dwarf phenotype. We have detailed the description of SD1-OE transgenic rice and slr1 gain-of-function mutant in Materials and Methods 4.1. (line316-319) It is known that Al tolerance was significantly higher in Japonica rice varieties than in Indica rice varieties (Yamaji et al, 2009, DOI: 10.1105/tpc.109.070771). The response of Nip and 9311 to Al stress has been investigated in the previous studies (Shu et al, 2015, DOI: 10.1016/j.rsci.2015.05.016; Roselló et al, 2015, DOI: 10.1016/j.jinorgbio.2015.08.021).

Reviewer 4 Report

Comments and Suggestions for Authors

The manuscript authored by Lu et al. examined the effects of gibberellic acids (GA) on aluminum (Al) toxicity, which has not been studied before. Initially, the expression of GA-related genes was assessed under Al stress. Subsequently, the Al tolerance/susceptibility of wild-type or GA-related mutants upon GA treatment was evaluated. The negative regulation of GA on Al resistance was clarified by its effects on reactive oxygen species (ROS) and the expression of Al tolerance genes. The manuscript is well-organized, and the writing is clear. I point out minor issues to improve the manuscript:

1. Line 37, “Currently, aluminum stress ranks as the second most severe stress on crops after drought”. This statement should be cautious because its validity depends on specific contexts such as time, location, types of stress (abiotic or biotic), and types of crops. The reference for this statement was published in 1995, and using more recent data is recommended.

2. Line 62, It's better to describe what the TaALM1 gene is, or at least provide the full name of this gene.

Results

3. Section 2.1, Line 106-108, The expression of GA-related genes should be presented meticulously and precisely. For example, SD1 expression increased at 6h, then reduced to the same level as the control after 24h. Meanwhile, GA2ox3, GA2ox6, GA2ox9, and SLR1 expression decreased to levels lower than the control. Additionally, comments on the expression of GA synthesis genes and the GA negative regulator gene, SLR1, are recommended, as their expressions showed a similar pattern.

4. Section 2.2, Line 121-122, Fig 2A, the authors investigated the impact of externally applied GA at various concentrations on rice growth under aluminum stress. Therefore, the images should depict how rice growth under Al treatment is affected by GA treatment. An additional control experiment (rice plant under Al treatment without GA application) is needed.

5. Figure 2, Figure 3, Figure 4, Relative root length (in the figure) and root relative growth rate are different parameters with different calculated formulas. Please clarify and maintain consistency throughout the whole manuscript. Besides, the authors did not describe how relative root length or root relative growth rate were evaluated in the Material and Methods. Please add this information. In the figure legend, does (n=6) represent the number of plants per assay, and was the experiment done only once? The number of biological replicas was not mentioned.

Comments on the Quality of English Language

Minor editing of English language required

Author Response

Line 37, “Currently, aluminum stress ranks as the second most severe stress on crops after drought”. This statement should be cautious because its validity depends on specific contexts such as time, location, types of stress (abiotic or biotic), and types of crops. The reference for this statement was published in 1995, and using more recent data is recommended.

Response: Thank you for your suggestion. We have deleted this sentence in the revised manuscript.

Line 62, It's better to describe what the TaALM1 gene is, or at least provide the full name of this gene.

Response: We have included the full name of TaALM1 gene in the revised manuscript (line 65-66).

Results

Section 2.1, Line 106-108, The expression of GA-related genes should be presented meticulously and precisely. For example, SD1 expression increased at 6h, then reduced to the same level as the control after 24h. Meanwhile, GA2ox3, GA2ox6, GA2ox9, and SLR1 expression decreased to levels lower than the control. Additionally, comments on the expression of GA synthesis genes and the GA negative regulator gene, SLR1, are recommended, as their expressions showed a similar pattern.

Response: Thank you for your suggestion. We have supplemented more detail description of GA-related gene expression patterns in the revised manuscript (line109-116). For the results of similar expression patterns of GA synthesis genes and GA negatively regulated genes, we can only speculate that GA may be involved in the response of rice to Al stress. Similarly, the jasmonate synthesis genes and signalling repressor genes, JAZs, are usually activated simultaneously under some kind of stress.

Section 2.2, Line 121-122, Fig 2A, the authors investigated the impact of externally applied GA at various concentrations on rice growth under aluminum stress. Therefore, the images should depict how rice growth under Al treatment is affected by GA treatment. An additional control experiment (rice plant under Al treatment without GA application) is needed.

Response: Effect of different concentration gradients of GA on rice growth under Al treatment was shown in Fig 2. Seedlings without GA application (0 μM GA) were used as the control to show the rice growth under normal and Al stress conditions (Fig 2).

Figure 2, Figure 3, Figure 4, Relative root length (in the figure) and root relative growth rate are different parameters with different calculated formulas. Please clarify and maintain consistency throughout the whole manuscript. Besides, the authors did not describe how relative root length or root relative growth rate were evaluated in the Material and Methods. Please add this information. In the figure legend, does (n=6) represent the number of plants per assay, and was the experiment done only once? The number of biological replicas was not mentioned

Response:

We have unified the description of relative root length in the main text (line129, 143, 156-157, 164, 175, 177, 187,329, 339, 383) and detailed the method of calculation in Material and Methods 4.2 (line329-331). Relative root length was calculated as the percentage of root length in the Al treatment relative to the root length in the non-Al treatment. For most of the experiments, representative results were shown in the manuscript after we independently repeated them at least three times to obtain similar results. In the figure legend, n = 6 means six biological replicates for each treatment in each experiment. We have added the description of the repetition of the experiments in the revised figure legends.